# Fiber Vector Bend Sensor Based on Multimode Interference and Image Tapping

**DOI:** 10.3390/s19020321

**Published:** 2019-01-15

**Authors:** Ziyang Zhang, Aashia Rahman, Julia Fiebrandt, Yu Wang, Kai Sun, Jiajun Luo, Kalaga Madhav, Martin M. Roth

**Affiliations:** 1School of Engineering, Westlake University, 18 Shilongshan Road, Hangzhou 310024, China; 2Institute of Advanced Technology, Westlake Institute for Advanced Study, 18 Shilongshan Road, Hangzhou 310024, China; 3innoFSPEC, Leibniz Institute for Astrophysics (AIP), an der Sternwarte 16, 14482 Potsdam, Germany; arahman@aip.de (A.R.); jfiebrandt@aip.de (J.F.); ywang@aip.de (Y.W.); ksun@aip.de (K.S.); jluo@aip.de (J.L.); kmadhav@aip.de (K.M.); mmroth@aip.de (M.M.R.); 4PicoQuant GmbH, Rudower Chaussee 29, 12489 Berlin, Germany

**Keywords:** fiber sensors, bend sensors, multimode interference, fiber imaging

## Abstract

A grating-less fiber vector bend sensor is demonstrated using a standard single mode fiber spliced to a multimode fiber as a multimode interference device. The ring-shaped light intensity distribution at the end of the multimode fiber is subject to a vector transition in response to the fiber bend. Instead of comprehensive imaging processing for the analysis, the image can be tapped out by a seven-core fiber spliced to the other end of the multimode fiber. The seven-core fiber is further guided to seven single mode fibers via a commercial fan-out device. By comparing the relative light intensities received at the seven outputs, both the bend radius and its direction can be determined. Experiment has shown that a slight bend displacement of 10 µm over a 1.2-cm-long multimode fiber in the X direction (bend angle of 0.382°) causes a distinctive power imbalance of 4.6 dB between two chosen outputs (numbered C4 and C7). For the same displacement in the Y direction, the power ratio between the previous two outputs C4 and C7 remains constant, while the imbalance between another pair (C3 and C4) rises significantly to 7.0 dB.

## 1. Introduction

Optical fibers are attractive candidates in sensing systems and offer many advantages over electrical and chemical sensors. The purely passive, sub-millimeter structure is immune to most electromagnetic interferences and electronic sparks. The fiber is also mechanically, thermally, and chemically stable to operate in harsh and critical environments [1,2]. For strain and structural deformation sensing, fiber Bragg gratings (FBGs) and long period fiber gratings (LPFGs) are popular choices owing to their simple structure, strong reflected signals, and potentially high sensitivity from a narrow spectral width [3,4]. Advanced three-dimensional (3D) sensors or shape sensors can be realized by writing orthogonally arranged FBGs along the multiple-cores of a common fiber, at different axial directions inside one fiber, or along separate fibers in one bundle [5,6,7]. Specialty fibers employing polygon-shaped cores [8,9], side-hole asymmetric fibers [10,11], and other novelty fibers [12] have also been exploited to build vector sensors, allowing the detection of stress, strain or deformation associated with the specific radial direction. Apart from their applications in safety monitoring, medical diagnosis and treatment, vector bend sensors may prove useful in telescope systems or other fine mechanics to track the angular position of the mirrors and to control the movement of the fiber-bundles.

The interrogation systems for the above mentioned vector sensors usually require spectrometer devices for spectral demodulation. Though integrated spectrometers such as arrayed waveguide gratings have been developed as compact FBG interrogators [13], the extra optical components can still add to the system complexity and cost. The solution of using distributed Rayleigh scattering eliminates the need to write gratings along the fiber and offers extremely high measurement resolution, but the interrogation system requires often complicated and expensive demodulation schemes involving optical frequency domain reflector [14,15]. An ideal vector bend sensor would consist of only standard, low-cost, and grating-less fiber-based components that are robustly spliced together. The demodulation system requires preferably only optical power detection, i.e., a small number of photo diodes (PDs) to bring sensor signal into the electric domain for further processing.

In this work, a vector bend sensor based on fiber multi-mode interference (MMI) and image tapping is proposed. The basic structure, shown in Figure 1, consists of a standard single mode fiber (SSMF, Corning SMF28 Ultra) center-spliced to a multimode fiber (MMF, FG105LCA) of the same cladding diameter. The MMI pattern can either be collected by a camera for comprehensive image processing, or in a simplified case, tapped out by an array of discrete fiber cores. In Figure 1, a multicore fiber (MCF), specifically a 7-core fiber (7CF, Fibercore SM-7C1500), is spliced at the end of the MMF. Subsequently a fan-out device (Fibercore, FAN-7C) is connected via a 7CF-connector. The seven output SMFs are fed to PDs. The power change among the seven outputs is evaluated to find out the image distortion due to disturbances along the fiber core, e.g., stress and curvature. It is worth mentioning that fiber fan-out devices have been widely used in astrophotonics as well as in optical communications with spatial division multiplexing technology to convert light propagation between SMFs and spatially multiplexed media (MCFs and MMFs) [16,17].

Previous efforts on fiber MMI sensors are mainly focused on the 1 × 1 MMI, where light in the MMF refocuses to a single spot. The application is mostly limited to scalar sensor, in which only the amplitude of the signal is detected by the intensity variation of the single imaging spot, while the orientation of the signal is lost [18,19,20]. In this work, the MMI length is chosen intentionally at a location away from the 1 × 1 self-focusing position. A well-spread, light intensity distribution is preferred, the variation of which is exploited to reveal vectorial information on the fiber curvature characteristics.

Other grating-less vector bend sensors have been reported using MCFs with asymmetrically placed cores that mix up the supermodes in the reflection spectrum. A small bending on the MCF can induce drastic changes in the supermodes, their excitation and in turn, a shift in the reflected spectrum [21,22]. Another variation adopts a short section of strongly coupled three-core fiber as the bend sensitive element and a subsequent mode-selective photonic lantern is used to regulate the bend-induced coupling to the output fibers [23], which requires no spectrometer for the interrogation. However, the design requires critical management and identification of each individual coupled mode. In this work, we generalize the situation and let the large number of modes in the MMF propagate freely. The demodulation relies on tapped-out, discrete image processing, eliminating the need to map each individual coupled mode.

## 2. Fiber MMI Design

The MMI structure consisting of the axis-aligned SMF and MMF is simulated using PhotonDesign (FIMMWAVE and BEAMPROP) software. The SMF has a core diameter of 8.2 µm and a numerical aperture (NA) of 0.14, while the MMF has a core diameter of 105 µm and a NA of 0.22. The cladding diameter of both fibers is 125 µm. Considering the large core size and relative large NA, mode selection techniques are used to limit the number of eigenmodes included in the simulation [24]. Upon entering the MMF, the single-mode launch light at 1550 nm starts to overlap/couple with the eigenmodes in the MMF and an intermodal interference develops along the fiber axis, as manifested by Figure 2a.

Since the structure is circularly symmetric with respect to the fiber axis, the interference pattern is accompanied by the self-imaging effects distributed in the radial plane in a concentric-ring pattern. At 10,670 µm, the 1 × 1 self-focusing appears. Though virtually any distributed imaging pattern can be used to analyze the bend-induced transition under thorough image processing, the actual pattern in this work is particularly chosen to facilitate the out-tapping via a 7CF. The pattern at 12,000 µm, shown in Figure 2b, offers a bright central spot as the “guide star” that can be coupled to the central core in the 7CF. At the same time a circle of minimal intensity is present with a radius of ~35 µm to the center, which is the same as the core pitch in the 7CF. This is beneficial in two aspects: (1) Good power contrast between the outer cores and the central core can be established in the initialization stage; and (2) power change among the outer cores can be easily detected when the fiber is bent.

To verify the simulation, the MMF is first spliced to the SSMF with a standard laboratory fiber fusion splicer and then subsequently cleaved at a distance of 12,000 µm using a standard fiber cleaver. A laser diode (LD) at 1550 nm is connected to the SSMF. The output of the MMI is focused onto an infrared (IR) camera (Princeton NIRVana 640ST) via an objective lens. The image, shown in Figure 2c, agrees well with the simulation results. The slight distortion of the experiment image, e.g., more pronounced central spot and slightly shifted light intensity concentration in the first ring toward the down-right direction, is attributed to the non-optimal positioning of the MMI, as the open end of the MMF to the camera makes it difficult to mount the fiber symmetrically without residue stress.

When the MMF is bent, both the propagation constants and the profiles of the eigenmodes will change. In PhotonDesign, the bend mode is calculated using the complex finite difference solver, which transforms the local cylindrical coordinates with respect to the imposed curvature in the mode-solver solutions. A one-degree bend has been implemented in the MMF and the simulated images at 12,000 µm are shown in Figure 3a,b, for the two orthogonal radial directions *X* and *Y*, respectively. The locations of the seven cores are also displayed to show the tap-out positions. The images in Figure 3 differ significantly from the unbent case in Figure 2b. According to the specifications, each core of the 7CF has a mode field diameter of 6.1 µm and a NA of 0.21. Mode overlap integral is performed to find out the coupling efficiency/transmission characteristics of each core to the specific MMI image at a certain bend angle and bend direction.

## 3. Bend Experiment

To characterize the variation in the tapped out light power experimentally, the 7CF is first marked on the outer cladding, indicating their relative core distribution and then spliced to the MMF. The other end of the 7CF is connected to the fan-out device via a MCF-connector. The SMF and the 7CF are placed in separate 3-axis fine stages, as shown in Figure 4a,b. The MMI/MMF is clamped at the two splicing points, as shown in Figure 4c. A metal pin is placed on a 3-axis fine stage and is used to pull the middle point of the MMF in both *X* and *Y* direction to induce curvature. The bending angle *θ*, the displacement of the pin *D* and the MMF length *L* (12,000 µm) are related by Equation (1). A displacement of 10 µm amounts to a bending angle of 0.382°.
(1)$$D=\frac{L}{2}\left(\frac{1}{\sin\theta /2}-\frac{1}{\tan\theta /2}\right)$$

The mounting and initialization of the fiber follows a three-step procedure. Firstly, the MCF is mounted with respect to the marker, so that the seven cores are labeled and oriented approximately, as indicated by the sketch in the middle of Figure 4d. The MMF and SMF are then carefully mounted, avoiding twists. Secondly, light is injected from the LD to the input SMF. The positions of the SMF and MCF are fine adjusted with respect to the measured power in the seven output SMFs. The target in this step is to find a uniform power distribution among the outer cores C2–C7, compensating any stress or curvature introduced during the mounting. Thirdly, the bend pin is pulled along the Y direction and the power change in core C4 and C7 are monitored. The fiber is rotated slightly until C4 and C7 undergo approximately the same power change. Step two and three have to be performed interactively to initialize the system.

The power measured from the LD directly to the PD is −4.34 dBm. Prior to the MCF splicing, the MMF is mounted in a ferrule/connector and fed into the PD. The power through the MMI is measured to be −4.66 dBm, and is used as reference. After splicing to the MCF and initializing the system, the MCF connector at the other end is fed to the PD and the power is measured to be −20.33 dBm. Finally, the power through the seven SMFs are measured and normalized to the reference. The results are summarized in Figure 4d, along with the simulated transmission from the MMI to the seven cores. The difference between simulation and measurement can be attributed to the loss of the fan-out device itself as well as the individual core/mode size variation among the seven cores. The insertion loss of the 1 m MCF + the fanout device is measured to be ~1 dB and the crosstalk between the cores is below −40 dB, both in good agreement with the product specifications.

To evaluate the sensitivity, the MMF is pulled from its initial unbend position in step of 10 µm in the *X* and *Y* direction, respectively. The power of the seven output SMFs are recorded and normalized to the reference accordingly. The transmission/coupling loss calculation is also performed by taking the modal integral between the distorted MMI image from the bent fiber and the individual single-mode profiles of the seven cores. The results are summarized in Figure 5, revealing a close agreement between the simulation and the measurement. Upon bending, the central spot of the MMI image gets “brighter” and more light is coupled to C1, as can be seen in both Figure 5a,b. When the MMI is bent in the *X* direction, the optical power received in C2 and C6 undergo approximately the same change, and the same goes for C3 and C5, attributing to the symmetry of the core layout. The power difference between C4 and C7 is mostly prominent. For a displacement of 10 µm, i.e., a bend angle of merely 0.328°, the power difference between C4 and C7 reaches 4.6 dB. When the MMI is bent in the *Y* direction, the outer cores form three pairs: C2/C3, C5/C6, and C4/C7. A large difference of 7.0 dB is observed between C3 and C5 when D reaches 10 µm. This difference diminishes gradually when *D* reaches 40 µm. However, the difference between C3 and C7 increases from −0.4 dB at *D* = 20 µm to 6.5 dB at *D* = 40 µm.

The dynamic bending range in this experiment is from 0 to 1.53° (*D* = 40 µm), which is larger than the reported value of 0.842° in Reference [21]. The experimentally demonstrated sensitivity is related to the 10-µm minimal step of the displacement. For the MMI length of 1.2 cm, the minimal displacement translates to a bending radius of 1.8 m, which can be well detected in both *X* and *Y* direction with large extinction ratios between the chosen outputs. Considering that the accuracy of the power meter is ±0.01 dBm, the proposed sensor is capable of detecting much weaker bend with bending radius well above 1.8 m, a value that is comparable to Reference [23].

## 4. Discussion and Conclusions

In this first demonstration, the detection scheme is simplified since the bend is oriented in the symmetry axis of the hexagonal core layout and the outputs form apparent pairs. In general cases, however, all seven powers may have to be analyzed and compared to find out the arbitrary bend direction and its amplitude. Additionally, noticed in Figure 5, is that the transition of the powers in the seven tapped out spots is not linear with respect to the displacement. A limited range of bend amplitude helps reduce the interrogation complexity. Detailed algorithms will be studied in the follow-up efforts. Though the fine stage has a movement resolution of 1 µm, the mechanical backlash and delay make it difficult to pin down the exact location at this accuracy. The uncertainty of the translational stage is around ±2 µm. In order to explore the detection limit, a piezo-controller is to be implemented in future work to regulate the fine movement of the bend pin in a more controllable and precise manner.

The MMI image forming is essentially a coherent effect. For long MMIs, a highly coherent laser source may be required to generate a sharp image and avoid the resolution drop due to blurred edges and smeared-out features. Multiple bending and twists along the MMI path will inevitably affect the intermodal interference condition and result in an altered image. It can be projected that image recognition in association with machine-learning approach can be used to train the sensor system for longer MMIs under complex fiber forms, as the conventional “bottom-up” approach relying on solving rigorous wave equations may prove too cumbersome for such large structures.

To summarize, a fiber vector bend sensor using MMI effect and image tapping technique is presented. The intermodal interference pattern in the MMF, its transmission characteristics upon bending and out-tapping properties via a 7CF and fan-out device are studied numerically and verified experimentally. The tapped out light power through the seven output SMFs shows distinctive transitions for the two orthogonal bend directions. Future work will be focused on the thorough investigation of the pattern transition with respect to arbitrary bending directions, as well as the effects of multiple bending and torsion.

In comparison to the grating-based fiber sensors, the proposed device is simple to fabricate: Three commercially available fibers, i.e., SSMF, MMF, and 7CF of the same outer cladding diameter is spliced together. Though in this work a fiber fan-out device is used to separate the 7 channels into individual SMF outputs, a simple PD array can be attached at the end-facet of the 7CF, or even the end-facet of the MMF if space allows, for direct power detection. No spectrometer is needed in the interrogation system. The bending information is extracted solely by comparing the optical power of the seven outputs. The easy-to-fabricate fiber sensor along with its simple interrogation method could be developed into compact sensor systems to be imbedded into fine-mechanics for on-site position tracking or wearable devices for health monitoring. The sensor may also prove useful in monitoring stress conditions at critical sites in building construction, in the crossbeam of bridges, at curved railway locations, etc.

## Figures and Tables

**Figure 1 sensors-19-00321-f001:**
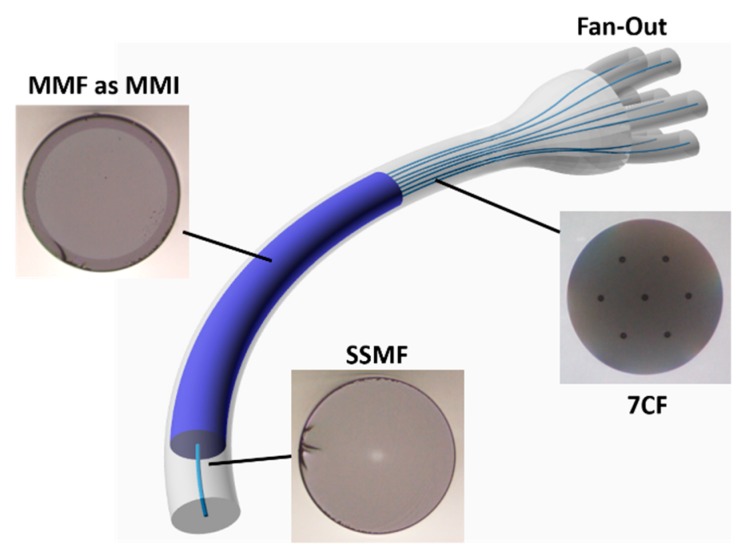
Layout of the vector bend sensor using a standard single mode fiber and a multimode fiber as MMI imaging device. The 7CF and the fan-out device are used to tap out the MMI image for analysis.

**Figure 2 sensors-19-00321-f002:**
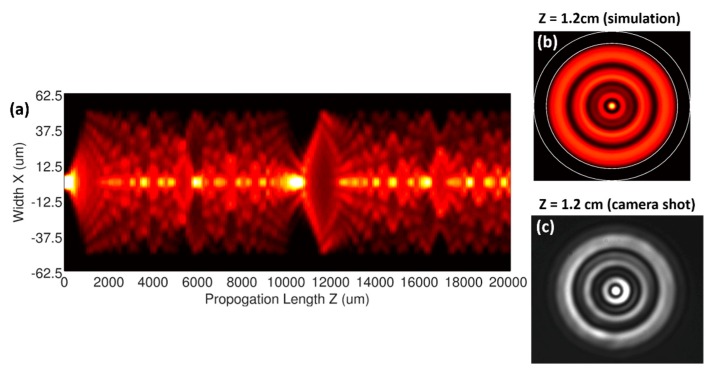
(**a**) Intermodal interference profile inside the MMF along the fiber axis. (**b**) Simulated light intensity distribution and (**c**) camera shot of the fiber MMI at 12,000 µm.

**Figure 3 sensors-19-00321-f003:**
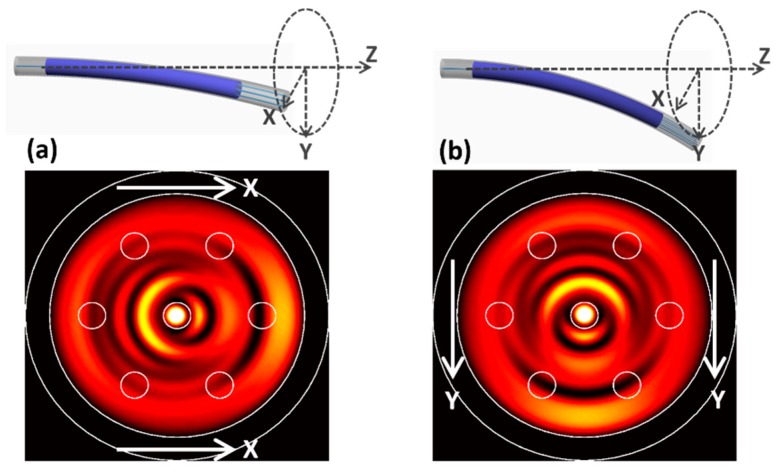
MMI image distortion induced by fiber bend in the (**a**) *X* and (**b**) *Y* direction.

**Figure 4 sensors-19-00321-f004:**
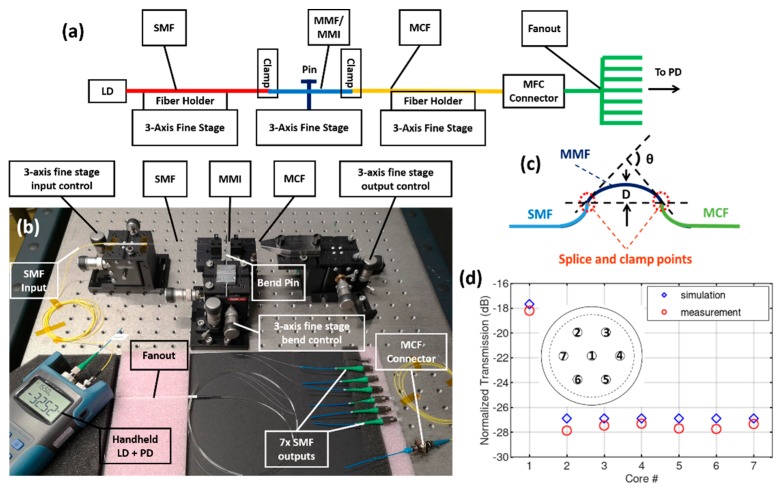
(**a**) Schematic layout and (**b**) photo of the experiment setup. (**c**) Mounting layout for the bent MMF/MMI. (**d**) Simulated and measured power distribution among the 7 output SMFs for the unbent case.

**Figure 5 sensors-19-00321-f005:**
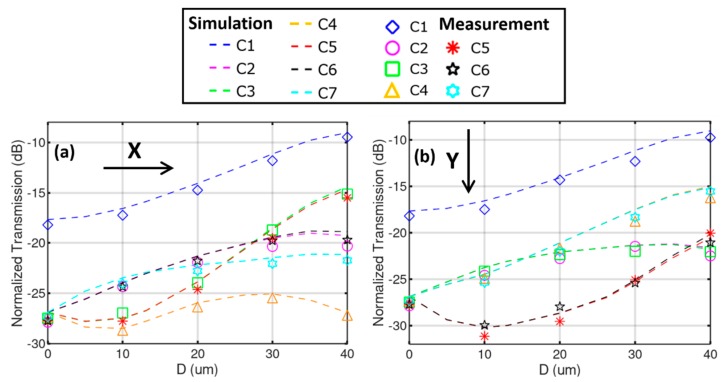
Simulated (dashed lines) and measured (markers) transmission in the 7 SMFs with respect to the displacement *D*, for bend in the (**a**) *X* and (**b**) *Y* direction, respectively.

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
