# Peer review of "Fiber Vector Bend Sensor Based on Multimode Interference and Image Tapping"

_sensors, 2019, doi:10.3390/s19020321_

Reviewer 1 Report

Authors present the design of a bend sensor based on a MMI and a 7 core PCF. Here authors used the 7 core PCF to tap the image after the MMI, avoiding the need of complex image processing. Moreover, according to authors the sensor can be able to determine in principle the direction, over x and y axes, of the bending. In general the manuscript is well organized and authors clearly described and supported the principle of operation of the sensing system. Therefore I would like to recommend the publication of the manuscript in Sensors after some minor revisions.

 Some specific aspects that I consider should be addressed by authors are the following.

1.       For clarity purposes, please add in figure 4 a sketch of your sensing setup, since in the current version it is difficult to appreciate each component from the photograph of your experiment.

2.       In figure 4c, you used a blue and a red lines to show the normalized transmission for each core of the PCF. Here, I found a bit confusing these continuous lines joining the core levels. Therefore, for clarity purposes I would like to suggest to represent these transmission levels in a discrete form (as the stem plot of Matlab) in such a way that we have a “stem” for each core.

3.       In the caption of figure 4, please describe each subfigure (a), (b) and (c).

4.       In figure 5, I recommend to change the color of the yellow lines since currently it is quite difficult to appreciate them

Author Response

Dear reviewer,

The reply to your comments have been summarized in the attached document.

Best Regards,

Authors.

Reviewer 2 Report

This manuscript describes a vector bend fiber sensor based on MMI effect, well known physical phenomenon for optical sensing. Even though the paper presents a novel fiber sensor that might be to the interested f fiber sensor research community, the technical presentation and interpretation of this paper need to be improved considerably prior to publication consideration.

Several comments are listed below hopefully will help to improve the manuscript in general:

1.     The abstract need to be polished so that it is more informative and self-contained so that any reader can know the significance of this paper and obtain sufficient information to decide if one should look into the main text. For example in “….between output C4 and C7. …”, the word C4 and C7 are not understandable unless on read further into the paper.

2.     The author should compare the performance of their sensor present in this work compared to other work by fellow researcher, such as the sensitivity, dynamic range, etc. One example is the following refencr: Joel Villatoro, Amy Van Newkirk, Enrique Antonio-Lopez, Joseba Zubia, Axel Schülzgen, and Rodrigo Amezcua-Correa, "Ultrasensitive vector bending sensor based on multicore optical fiber," Opt. Lett. 41, 832-835 (2016)

3.     The specification of the multicores fiber need to be present in detail or the author should characterize some important parameter such as the cross talk between the cores. This is important as the power ratio in core #1 is much higher than the rest.

4.     The uncertainty of the experiment apparatus such as the translational stage and power meter need to be described in detail.

5.     The experiment result of MMI image distortion induced by fiber bend in both X and Y direction which are simulated in Figure 3(a) and 3(b) is crucial to support author claim for this paper. This will be a solid proof of the physical principle of this sensor, which is later complimented by the quantifying result from the power ratio measurement.

6.     Some potential applications should be highlighted, especially in relevant with the current and potentially better benchmark that can achieved by current sensor design.

Author Response

Dear reviewer,

The reply to your comments has been summarized in the attached document.

Best Regards,

Authors.

Round  2

Reviewer 2 Report

The author's replies and revised manuscript are satisfactory. The improved manuscript is recommended for publication in Sensor.